# Study into the Fire and Explosion Characteristics of Polymer Powders Used in Engineering Production Technologies

**DOI:** 10.3390/polym15214203

**Published:** 2023-10-24

**Authors:** Richard Kuracina, Zuzana Szabová, Eva Buranská, László Kosár, Peter Rantuch, Lenka Blinová, Dagmar Měřínská, Peter Gogola, František Jurina

**Affiliations:** 1Institute of Integral Safety in Trnava, Faculty of Materials Science and Technology, Slovak University of Technology in Bratislava, Ul. Jána Bottu 2781/25, SK-917 24 Trnava, Slovakia; eva.buranska@stuba.sk (E.B.); laszlo.kosar@stuba.sk (L.K.); peter.rantuch@stuba.sk (P.R.); lenka.blinova@stuba.sk (L.B.); 2Department of Production Engineering, Faculty of Technology, Tomas Bata University in Zlín, Vavrečkova 5669, CZ-760 01 Zlín, Czech Republic; merinska@utb.cz; 3Institute of Materials in Trnava, Faculty of Materials Science and Technology, Slovak University of Technology in Bratislava, Ul. Jána Bottu 2781/25, SK-917 24 Trnava, Slovakia; peter.gogola@stuba.sk; 4Institute of Production Technologies in Trnava, Faculty of Materials Science and Technology, Slovak University of Technology in Bratislava, Ul. Jána Bottu 2781/25, SK-917 24 Trnava, Slovakia; frantisek.jurina@stuba.sk

**Keywords:** polyamide, polypropylene, UHMW polyethylene, dust explosion, hazard

## Abstract

Polymers and their processing by engineering production technologies (injection, molding or additive manufacturing) are increasingly being used. Polymers used in engineering production technologies are constantly being developed and their properties are being improved. Granulometry, X-ray, FTIR and TGA were used to characterize polymer samples. Determination of the fire parameters of powder samples of polyamide (PA) 12, polypropylene, and ultra-high molecular weight (UHMW) polyethylene is the subject of the current article. An explosive atmosphere can be created by the powder form of these polymer materials, and introduction of preventive safeguards to ensure safety is required for their use. Although the fire parameters of these basic types of polymers are available in databases (e.g., GESTIS-DustEx), our results showed that one of the samples used (polypropylene) was not flammable and thus is safe for use in terms of explosiveness. Two samples were flammable and explosive. The lower explosive limit was 30 g·m^−3^ (PA12) and 60 g·m^−3^ (UHMW polyethylene). The maximum explosion pressure of the samples was 6.47 (UHMW polyethylene) and 6.76 bar (PA12). The explosion constant, K_st_, of the samples was 116.6 bar·m·s^−1^ (PA12) and 97.1 bar·m·s^−1^ (UHMW polyethylene). Therefore, when using polymers in production technologies, it is necessary to know their fire parameters, and to design effective explosion prevention (e.g., ventilation, explosive-proof material, etc.) measures for flammable and explosive polymers.

## 1. Introduction

Owing to their enhanced design flexibility and cost-effective manufacturing solutions, utilization of polymer materials in engineering production technologies, such as injection molding and additive manufacturing, has brought about significant advancements across diverse industries [1,2,3].

However, concerns have been raised regarding the potential fire and explosion hazards associated with the increasing use of polymer powders in these processes.

Common flammable dust accidents are causing industrial process managers to focus on flammable powder properties and safer designs. Underestimating risks during safety analyses is often due to a lack of understanding and insufficient analysis of characteristic explosion parameters. The absence of experimental data poses a risk of underestimating process hazard and overestimating safety measures [4,5,6].

The development and use of modified polymers is increasingly common in industry. Polymers used in engineering production technologies are composed of the majority of components (polyethylene, polypropylene, polyamides, polystyrene, etc.) which are modified with additives. These additives can significantly change the mechanical and physical properties of polymers (e.g., strength, melting point, etc.) as well as their fire properties. The polymer can either have higher fire parameters or become non-explosive. An increase in fire parameters can be caused by addition of dyes or flammable organic fillers, for example. A decrease in fire parameters can be caused by inorganic fillers (e.g., salts, oxides, etc.), antioxidants (e.g., phenol based aromatic amines) or fire retardants (metal hydroxides, halogen and phosphorous-based compounds, etc.). Different types of additives in polymers have been discussed by various authors [7,8,9]. There is little information in scientific articles dealing with the measurement and evaluation of the explosion parameters of polymers [10]. The influence of additives on the fire parameters of the polymer must always be verified by laboratory measurements.

Preventing explosion hazards involves knowing the characteristics of dust parameters, such as the minimum ignition energy (MIE), the maximum rate of pressure rise (explosion constant—K_st_), the dust minimum oxygen concentration (MOC), the maximum pressure (P_max_), the minimum ignition temperature of the dust layer (MITL), the minimum ignition temperature of the dust cloud (MITC) or the limiting oxygen concentration (LOC) [6,11,12].

The explosion parameters in this study are measured according to the EN 14034 standard in a spherical explosion chamber [13]. During the measurement, pressure changes are recorded at a rate of at least 1000/s. Ignition of the dispersed cloud of the sample is ensured by an igniter with an energy of 2 × 5 kJ. The ignition temperature of dispersed dust from a hot surface is determined in a G-G furnace according to EN ISO/IEC 80079-20-2 [14]. The ignition of the dust after its dispersion in the heated tube is visually observed. The authors also deal with determining and assessing the fire parameters of different types of samples under different conditions [15,16].

The aim of the research was to investigate and analyze the fire and explosion characteristics of various polymer powders commonly used in engineering production technologies. Thanks to conduction of comprehensive experimental studies, this paper provides valuable insight into the behavior of polymer powders under different conditions and concentrations, while shedding light on potential risks and safety implications in industrial settings.

Understanding the explosion characteristics is vital for preventing dust explosions. Measuring the explosion severity and ignition sensitivity parameters of polymer dust is of utmost importance for enterprises in order to prevent and mitigate potential polymer dust explosions.

## 2. Materials and Methods

Three samples of polymers used in production technologies were tested—PA12 (polyamide), UMHW PE (polyethylene) and PP (polypropylene). The samples were not treated before the measurement. The characterization of their properties (granulometry, LSM, SEM, X-ray and FTIR) is presented in the following sections of the presented article.

### 2.1. Polyamide (PA) 12

The study investigated polyamide 12 by VESTOSINT^®^ X7004 (trade name), a polymer polyamide 12 powder utilized for injection-molded parts and high-quality sintered coatings. According to the Material Safety Data Sheet (MSDS), the melting point of PA12 is 180 °C [17,18].

Thermogravimetric analysis [19] demonstrated that the melting point of PA12 is 185 °C, with the decomposition of its chains commencing at a temperature of 325 °C.

In our previous study, the polymer powder PA12 Sinterit utilized in laser sintering was found to be known for exhibiting explosive characteristics. The highest explosion overpressure recorded for the sample was 6.78 bar at a concentration of 750 g·m^−3^ [20].

Another study focused on the incendiary powder PA12 with a bimodal particle size distribution (10 μm and 55 μm). It was revealed that the minimum ignition energy of PA12 is below 40 mJ for concentrations approaching 1000 g·m^−3^ [11].

In the field of laser sintering, semi-crystalline thermoplastics, particularly polyamides (PA12 and PA11), are the most commonly utilized polymers. These polyamides account for over 95% of the available powder used in the polymer additive manufacturing market within the industry [21,22].

### 2.2. Polypropylene (PP)

For the measurements, natural polypropylene from Borealis (Vienna, Austria) known as BorPlus SE523MO was utilized. This particular grade of polypropylene is intended for rotational molding applications and has been specially designed to improve impact performance, particularly in low-temperature conditions [23,24].

Polypropylene (PP), a type of polymer, is widely utilized in various industries, including transportation, furniture, automotive, insulation, electronics, electric casings, interior decorations, and architectural materials [25,26].

Natural polypropylene is an organic substance that is flammable. The influence of the additive ammonium polyphosphate on flammability and explosiveness was discussed by the authors in [26]. The thermal decomposition temperature of PP in the air atmosphere was found to be 250 °C. The particle size distribution was observed to reach d(10) = 8.64 µm, d(50) = 29.2 µm, and d(90) = 128 µm. The maximum values of P_max_ and K_st_ were measured at 8 bar and 257 bar·m·s^−1^, respectively. P_max_ and K_st_ exhibited an initial increase with the rise in dust concentration, followed by a decrease. Furthermore, the minimum explosible concentration (MEC) of PP powders was determined to be 25 g·m^−3^ [26]. The melting point of PP was in the range of 150–166 °C, flash point >300 °C [27,28].

### 2.3. Ultra-High Molecular Weight Polyethylene (UHMW-PE)

For the experiment, a powder of ultra-high molecular weight polyethylene (UHMW-PE), known as GUR^®^ 2024-PE-UHMW and produced by Celanese (Irving, TX, USA) was used. This material demonstrates a significantly higher molecular weight compared to standard PE and exhibits a density of 930 kg/m^3^. UHMW-PE is known for its high wear resistance, high toughness, high impact strength and low friction coefficient, as well as durability, biocompatibility and chemical inertness and excellent mechanical characteristics, even in cryogenic conditions. Its MFR (melt flow rate) temperature is 190 °C, and it can be processed through compression molding and film extrusion techniques [29].

The melting point of UHMW-PE was found to be around 136 °C, and the decomposition of its chains begins at a temperature of 429 °C, as shown by thermogravimetric analysis [30].

Typically, the UHMW-PE powder, as it is after polymerization, undergoes processing through ram extrusion or compression molding techniques at elevated temperatures (200–240 °C) and high pressures (8–10 MPa) [31]. These processes are intricate, costly and time-consuming, often requiring several hours to complete. In 2007, a new approach to UHMW-PE processing was introduced, known as impact compaction [32]. This method involves applying a series of blows from an impactor to the powder enclosed in a metal mold, resulting in cyclic impact compaction. This technology allows for the rapid production of small, flat-shaped parts in just a few minutes.

### 2.4. Granulometry, Topography (LSM, SEM), X-ray, FTIR and TGA of the Samples

These techniques are commonly used in materials science and research to analyze the properties, composition and behavior of various materials. They provide valuable insight into the structure and characteristics of materials at different levels of detail.

#### 2.4.1. Granulometry

The particle size distribution of the polyamide 12, polypropylene, and UHMW polyethylene samples was determined using sieve analysis. This analysis procedure followed the EN 933-1 Standard [33]. Sieve analysis was carried out using a Retsch AS 200 sieving machine (Retsch GmbH, Haan, Germany) for 15 min with an amplitude of 2 mm/G. The results of the sieve analysis, including the median values, are presented in Table 1. Figure 1 and Figure 2 show the particle shapes of the polyamide 12, polypropylene, and UHMW polyethylene powders.

#### 2.4.2. Topography

The surface characteristics of powder particles were recorded using a ZEISS LSM700 (Carl Zeiss AG, Oberkochen, Germany) scanning confocal microscope. A 405 nm light source was employed, and when combined with an Epi-plan-Apochromat 100×/0.95 objective, it allowed for achieving resolution step sizes of 110 nm along the X and Y axes and 60 nm along the Z axis, as depicted in Figure 1 and Figure 2.

#### 2.4.3. X-ray

Figure 3 shows the XRD patterns recorded for each polymer powder investigated. The X-ray diffraction measurements were performed using a PANalytical Empyrean diffractometer (Malvern Panalytical Ltd., Malvern, UK) with a Ni-filtered Cu–Kα radiation. XRD patterns were recorded in the range of 10–110° 2Theta.

Figure 3a shows the XRD pattern for the investigated polyamide 12 powder. The obtained XRD pattern confirms the presence of the crystalline γ phase (peaks at 11.2° and 21.5° 2Theta); however, basically no peaks of the α phase could be identified [34] (PDF 00-057-1433) [35,36,37,38,39].

The presence of crystalline polypropylene can be confirmed for Sample 2, based on Figure 3b. All peaks can be related to the isotactic α form of polypropylene, based on the JCPDS ICCDD database PDF 00-061-1416. This is in line with multiple publications [40,41,42,43]. For polypropylene, a β form is also reported to form under certain conditions [44,45]. However, this was not found in our investigated sample.

The XRD pattern of the UHMW polyethylene is shown in Figure 3c. A very good overlap with multiple publications investigating UHMW polyethylene was found [31,46,47]. Interestingly, the diffraction pattern of UHMW polyethylene matches the one of HD-PE [41], JCPDS ICCDD database PDF 00-060-0986).

#### 2.4.4. FTIR

The ATR-FTIR spectra were obtained using a Varian FT-IR Spectrometer 660 from Agilent Technologies, Inc., located in Santa Clara, CA, USA. The specimens were directly placed on a diamond crystal of the ATR accessory known as GladiATR, manufactured by PIKE Technology Inc. in Madison, WI, USA. The resulting spectra underwent correction to account for air absorbance in the background. These spectra were recorded utilizing a Varian Resolutions Pro instrument and involved measurements within the 4000–400 cm^−1^ range. Each spectrum was acquired 256 times at a resolution setting of 4, as illustrated in Figure 4.

Figure 4c demonstrates the infrared spectrum of UHMW polyethylene with absorption bands which correspond to the following chemical structure—2911 cm^−1^ (CH_2_ asymmetric stretching vibrations), 2845 cm^−1^ (CH_2_ symmetric stretching vibrations), 1460 cm^−1^ (CH_2_ bending vibrations) and 716 cm^−1^ (C–CH_2_ rocking vibrations) [48,49].

Figure 4b illustrates the spectrum of PP. Absorption bands located at specific wavenumbers correspond to the following functional groups—2948 cm^−1^ (CH_3_ asymmetric stretching vibrations), 2915 cm^−1^ (CH_2_ asymmetric stretching vibrations), 2870 cm^−1^ (CH_3_ symmetric stretching vibrations), 2837 cm^−1^ (CH_2_ symmetric stretching vibrations), 1451 cm^−1^ (CH_3_ symmetric bending vibrations), 1374 cm^−1^ (CH_3_ umbrella mode), 1164 cm^−1^ (C–H wagging vibrations, CH_3_ rocking vibrations), 995 cm^−1^ (CH_3_ rocking vibrations, C–C stretching vibrations), 970 cm^−1^ (CH_3_ rocking vibrations, C–C stretching vibrations), 838 cm^−1^ (C–H rocking vibrations) and 811 cm^−1^ (C–C stretching vibrations) [50,51].

The spectrum of polyamide 12 (Figure 4c) shows the peaks at the indicated wavenumbers which correspond to the following functional groups—3280 cm^−1^ (N–H stretching vibrations), 3083 cm^−1^ (overtone of N–H bend), 2914 cm^−1^ (CH_2_ asymmetric stretching vibrations), 2846 cm^−1^ (CH_2_ symmetric stretching vibrations), 1633 cm^−1^ (Amide I, C=O stretching vibrations), 1542 cm^−1^ (Amide II, N–H in plane bending vibrations), 1460 cm^−1^ + 1435 cm^−1^ + 1364 cm^−1^ (C–H bending vibrations), 1263 cm^−1^ (Amide III, C–N stretch) and 718 cm^−1^ (N–H out-of-plane bending vibrations) [52,53,54].

### 2.5. MIT of Dispersed Dust

The measurement of minimum ignition temperature (MIT) for dispersed dust was carried out using standardized equipment known as the Godbert–Greenwald furnace, as depicted in Figure 5. This furnace is specifically designed to determine the MIT of dispersed dust particles. In this test, small quantities of dust are directed vertically downward through a heated furnace, and any ignition is detected through visual inspection.

The test material is introduced into the furnace using an air blast for dispersion. It is worth noting that since the majority of the sample comprises two fractions (>32 μm, >45 μm), the resulting MIT value is primarily influenced by these two fractions. Other fractions, due to their lower percentage in the sample, have a negligible impact on the MIT value of the dispersed dust when it is in contact with a hot surface.

The amount of dust used for testing is 0.15 g, which corresponds to the concentration associated with the highest P_max_ value. The dust is dispersed under air pressures of 20 kPa and 50 kPa. Ignition is recognized if a burst of flame is observed below the end of the furnace tube. For each combination of temperature and pressure, five measurements were conducted, and the measurements were considered positive (“YES”) if at least one test yielded ignition.

The MIT value for the dispersed dust is determined as the lowest furnace temperature at which ignition occurred, with a 20 K subtraction, according to the defined procedure [14]. Due to the characteristics of the dust as indicated in the Material Safety Data Sheet (MSDS), specifically its melting point of 180 °C, MIT testing for settled dust was not performed.

### 2.6. Explosion Parameters

The explosion parameters of the samples were determined using the KV 150M2 explosion chamber (OZM Research, Hrochův Týnec, Czech Republic) (Figure 6 and Figure 7). Compressed air from a pressurized vessel (6.5 L at 10 bar) was used to disperse the dust. The chamber has a volume of 365 L.

To perform the test, the sample was placed on a disperser plate and dispersed using a stream of compressed air. Subsequently, the sample was ignited using a pyrotechnic igniter with an energy of 2 × 5 kJ. The igniter was centrally located within the explosion chamber in accordance with EN 14034 Standard [13]. There was a 350 ms delay between opening the dispersing valve and activating the igniter.

Pressure changes within the chamber during the dust cloud explosion were recorded using pressure transducers. These measurements were recorded at a rate of 50,000/s. Pressure changes were recorded at various dust concentrations. Each concentration was tested three times, and the highest value obtained during these measurements was recorded.

## 3. Results and Discussion

Thermogravimetric analysis (TGA) was performed using a NETZSCH STA 449 F5 Jupiter analyzer. TGA and DSC (differential scanning calorimetry) were used for measuring the energy change of the polymer by increasing the temperature in air (Figure 8) and nitrogen (Figure 9) atmosphere. Aluminum cells were used for analysis of the polymers. The temperatures of the sample cells were increased by 10 K/min. The weights of all of the samples were 10.0 ± 1.0 mg. The measurements were performed in a stream of pure nitrogen and a mixture simulating air (80%_vol._ N_2_ and 20%_vol._ O_2_). The gas flow rate was 100 mL·min^−1^. TGA results can be affected by sample weight [58]. TGA was not used in this research to study pyrolysis kinetics or thermo-oxidation of polymers. The weight of the sample was chosen based on commonly used values [59,60,61]. The results are shown in Table 2 and Table 3.

Decomposition of polymer samples in nitrogen occurred in one step. The initial mass loss temperature of polyamide 12 was 126 °C, polypropylene was 221 °C and UMHW polyethylene was 287 °C. The undecomposed residue at 650 °C was about 1% of the weight of the samples.

In the air, the behavior of individual polymers was different. The weight loss was due to the presence of oxygen at lower temperatures. Polypropylene weight loss took place in two stages. The main decrease (oxidation reactions) was observed at a temperature of 342 °C (1st peak). The thermogravimetric curve of polypropylene in the air is very similar to the curve obtained by the measurement in nitrogen.

Decomposition of polyamide 12 was observed at higher temperatures. The course was similar to polypropylene. However, the main decrease was at a temperature of 439 °C (2nd peak). The decrease caused by oxidation reactions (1st peak) was recorded at a temperature of 359 °C.

UHMW polyethylene decomposed in oxygen in several steps. First, (Peak 1) at temperatures around 200 °C, there was an increase in the weight of the sample (caused by binding the oxygen from the surrounding atmosphere). Three more stages of decomposition followed. The main loss (Peak 3) was observed at a temperature of 425 °C (−70% weight loss). Similar to the measurements in nitrogen, the undecomposed residue after the measurements was lower than 2% in all cases.

The measured values of the explosion parameters in the KV-150M2 chamber are shown in Table 4 and Figure 10, Figure 11, Figure 12, Figure 13, Figure 14 and Figure 15.

The measured values show that two samples are flammable. The highest values of explosion parameters were achieved with the polyamide 12 sample. The maximum explosion pressure P_max_ was 6.76 bar at a concentration of 750 g·m^−3^. The highest rate of pressure rise was at a concentration of 500 g·m^−3^ with a value of 163.2 bar·s^−1^ (K_st_ = 116.6 bar·m·s^−1^). The lower explosive limit of the sample was 30 g·m^−3^.

For the UHMW polyethylene sample, the highest values were recorded at a concentration of 500 g·m^−3^. The explosion pressure value P_max_ was 6.47 bar and the rate of pressure rise was 135.9 bar·s^−1^ (K_st_ = 97.1 bar·m·s^−1^). The lower explosive limit of the sample was 60 g·m^−3^. The same time response to the ignition source was observed for both flammable polymer samples.

No explosion was observed in the polypropylene sample at any of the measured concentrations. Despite the fact that polypropylene is a flammable substance, the additives contained in this polymer make it non-flammable. In practice, inorganic substances (oxides, carbonates, phosphates, generally various salts, etc.) or special organic substances (antioxidants) can be used as additives.

The results (Table 5 and Table 6) of the measurements in accordance with the EN ISO/IEC 80079 Standard [14] show that the MIT of dispersed dust for polyamide 12 is 350 °C and the MIT for the UHMW polyethylene sample is 320 °C. Polypropylene did not ignite when measuring the MIT dispersed dust. The use of polypropylene material is safe in the entire temperature range (up to 450 °C). Polyamide 12 and UHMW polyethylene polymers can be safely used at temperatures up to 300 °C.

## 4. Conclusions

The fire parameters of three polymer samples used in production technologies were determined. Currently, production technologies work with polymers more and more frequent.

Several methods can be used to characterize samples and their chemical composition. The polymers presented in this scientific study are composed of several components (polymer + additives) and X-ray, FTIR, and TGA methods were used for their characterization. Polymers have a high content of the combustible majority component (polymer). They also contain a small content of additives that affect fire parameters.

They often use powdery materials that can be flammable and explosive. Despite the fact that the basic component of such materials are polymers, various modifications and additives can significantly change their fire properties. Based on the measurement of fire parameters, it is then possible to assess the safety of their use.

The fire parameters of three types of polymers were compared in the research described in this article. The results allow us to conclude that two of the three samples are flammable and explosive.

The LEL values are 30 and 60 g·m^−3^, the maximum explosion pressure is more than 6 bar g and the explosion constant is 100–135 bar·m·s^−1^. The minimum ignition temperature of dispersed dust from a hot surface is in the range of 320–350 °C. Therefore, when using these materials, it is necessary to apply the principles of explosion protection for their safe use. The polypropylene sample was evaluated as non-flammable and non-explosive. It contains additives that make it non-flammable. Polypropylene use is safe, even when it comes to dispersion and contact with a hot surface. It can be concluded that the recommended way of increasing the fire safety level of polymers is by use of suitable additives. This can significantly increase the fire safety of polymer materials.

From the results, it can be concluded that the polymers used in additive engineering technologies can be flammable or non-flammable. The non-flammability of polymers is usually achieved by additives. The use of such polymers is then safe from the point of view of fire risk.

However, the majority of polymers used in industry have a significant fire risk. Therefore, it is important to focus on these flammable polymers. Fire parameters are basic data necessary for the design of effective explosion prevention measures (e.g., ventilation or explosion vent relief). The specific design of explosion prevention is always a combination of fire parameters of polymers and design and layout parameters in the production company.

The development of polymers in the field of additive engineering technology is still progressing. New types of additives are being developed for polymers. Therefore, in the future it is possible to focus on determining the fire parameters of such new types of polymers. Studies will focus on polymers and their additives, which can increase the level of safety in industrial operations in the future.

## Figures and Tables

**Figure 1 polymers-15-04203-f001:**
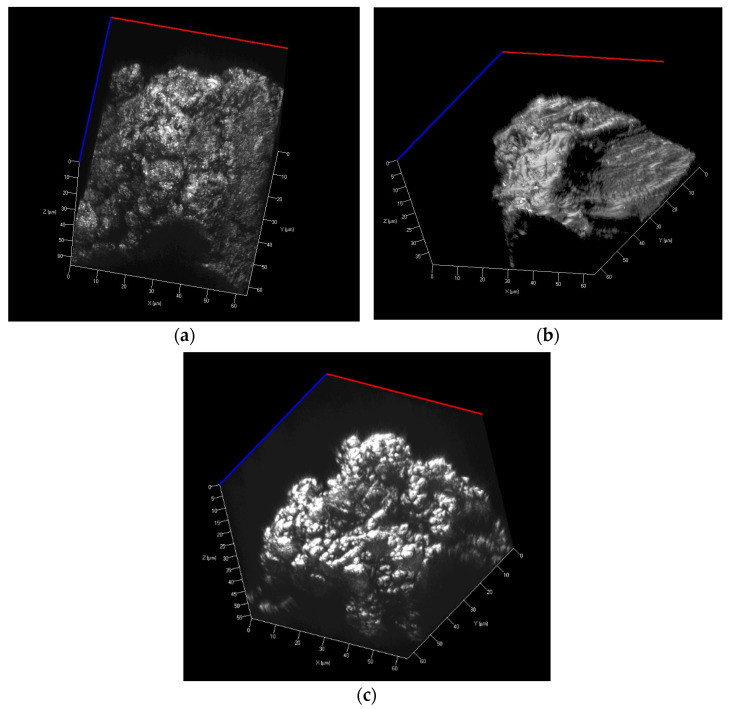
Confocal LSM (laser scanning microscope) images of particles of (**a**) polyamide 12, (**b**) polypropylene, and (**c**) UHMW polyethylene.

**Figure 2 polymers-15-04203-f002:**
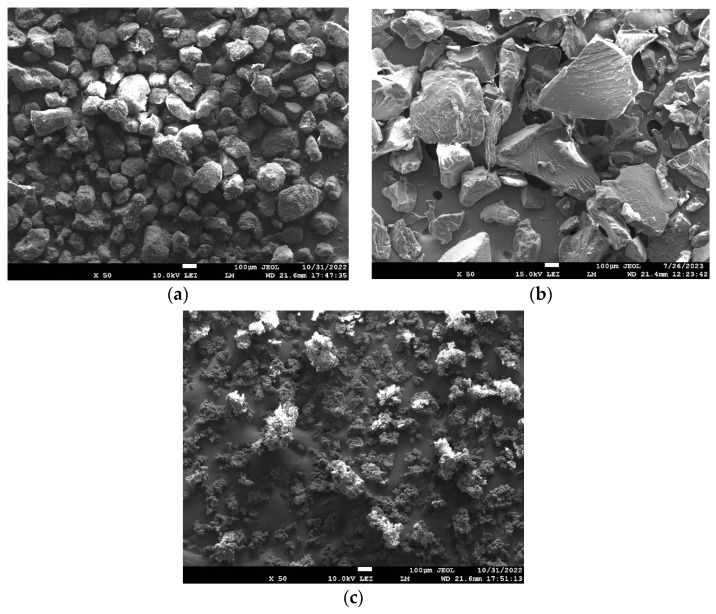
SEM (scanning electron microscope) images of fracture surfaces of (**a**) polyamide 12, (**b**) polypropylene, and (**c**) UHMW polyethylene.

**Figure 3 polymers-15-04203-f003:**
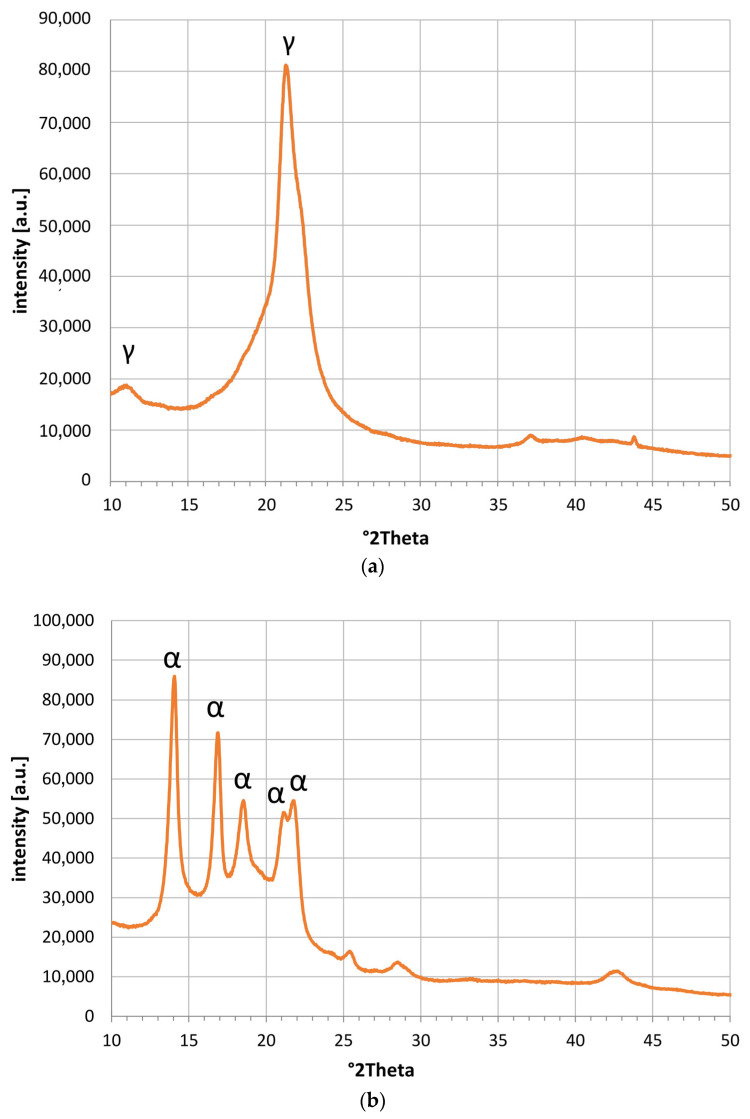
X-rays of (**a**) polyamide 12, (**b**) polypropylene, and (**c**) UHMW polyethylene.

**Figure 4 polymers-15-04203-f004:**
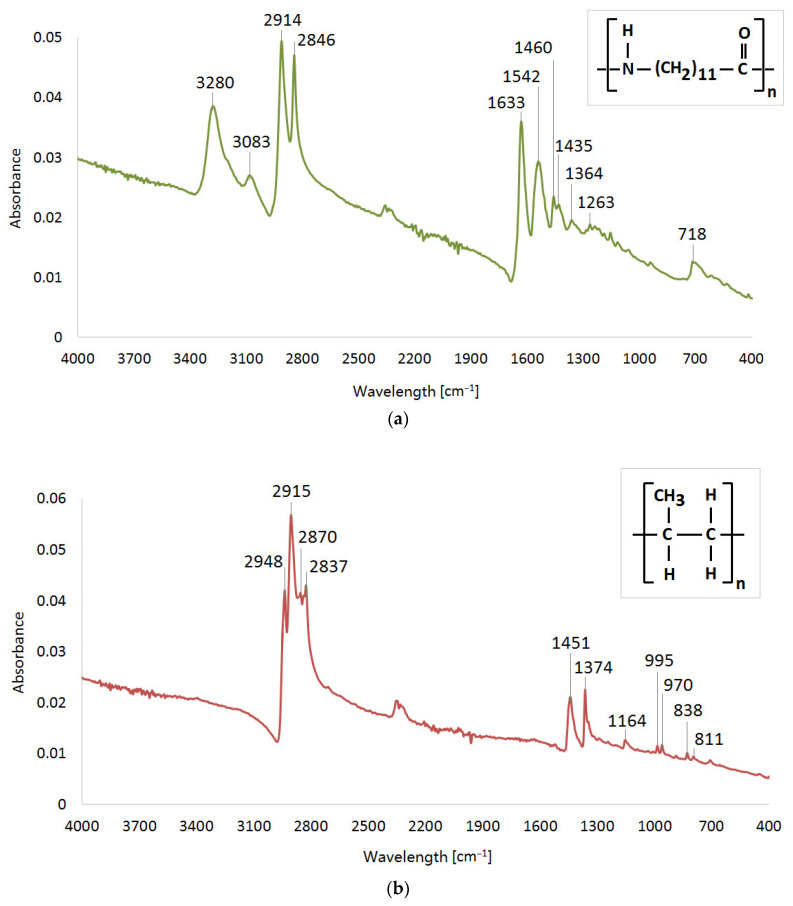
Infrared spectra of (**a**) polyamide 12, (**b**) polypropylene, and (**c**) UHMW polyethylene.

**Figure 5 polymers-15-04203-f005:**
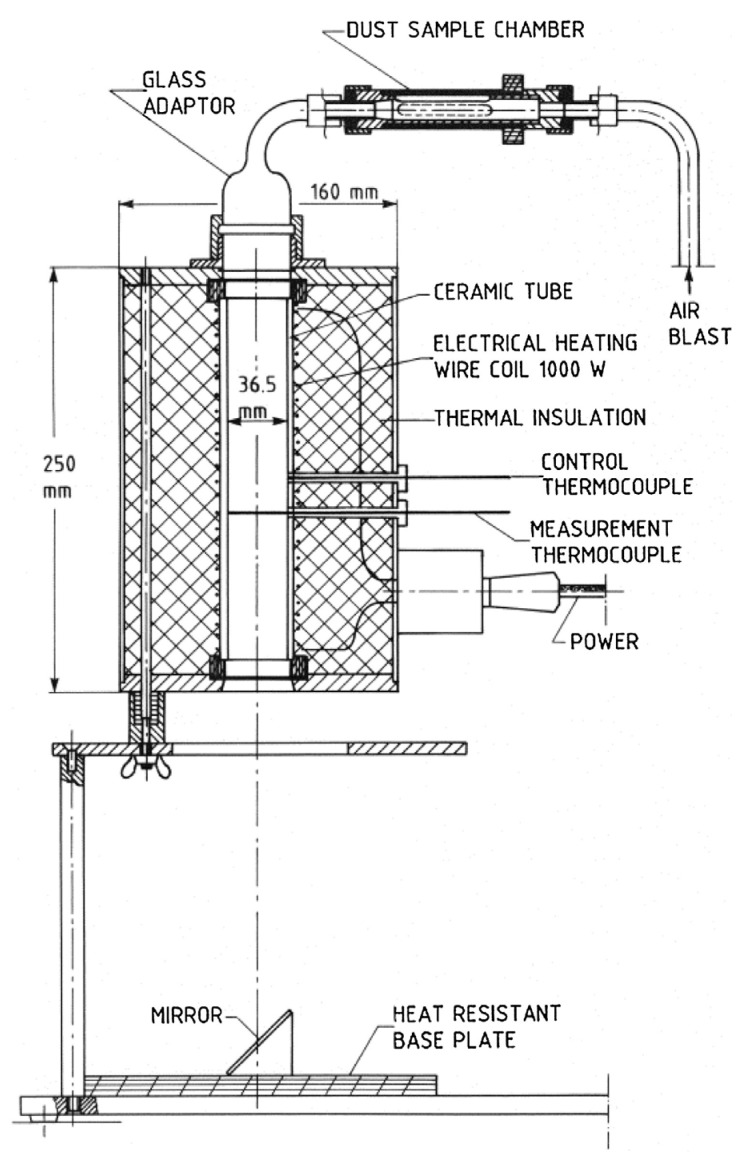
Cross-section of Godbert–Greenwald furnace [55].

**Figure 6 polymers-15-04203-f006:**
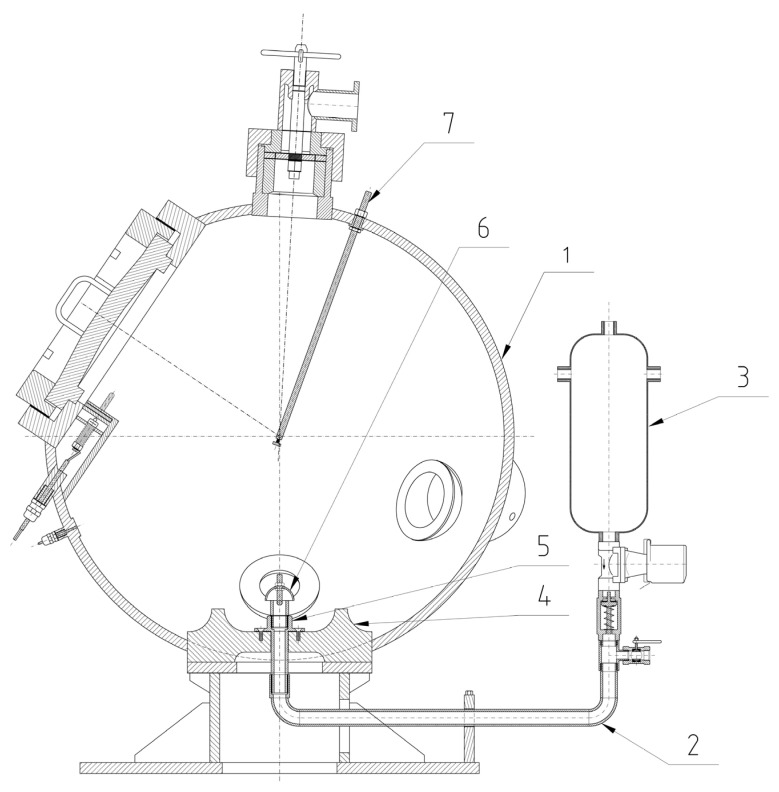
Cross-section of the KV 150M2 explosion chamber [56] (1, chamber; 2, disperser tube; 3, air pressure vessel; 4, dispersing plate; 5, disperser; 6, air flow reverser; 7, metal rod for supply the igniter).

**Figure 7 polymers-15-04203-f007:**
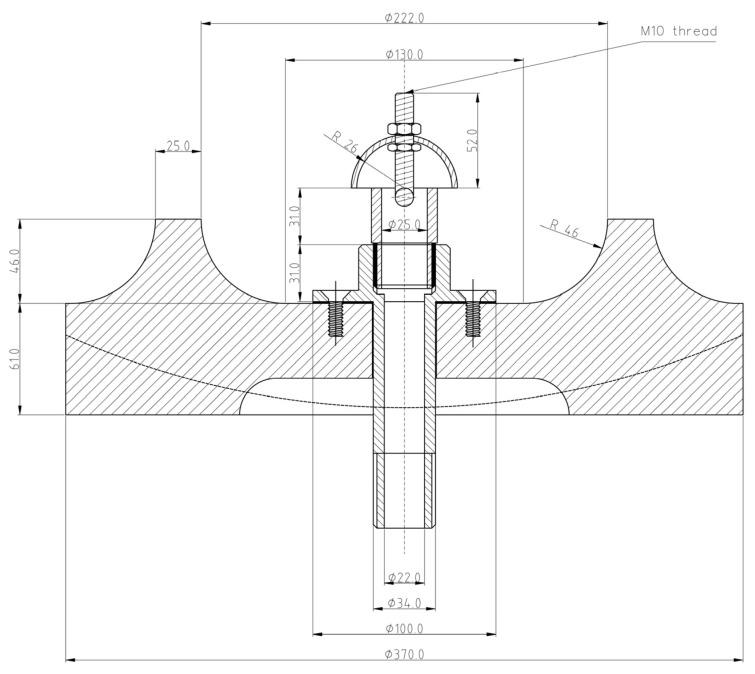
Cross-section of disperser (unit: mm) [57].

**Figure 8 polymers-15-04203-f008:**
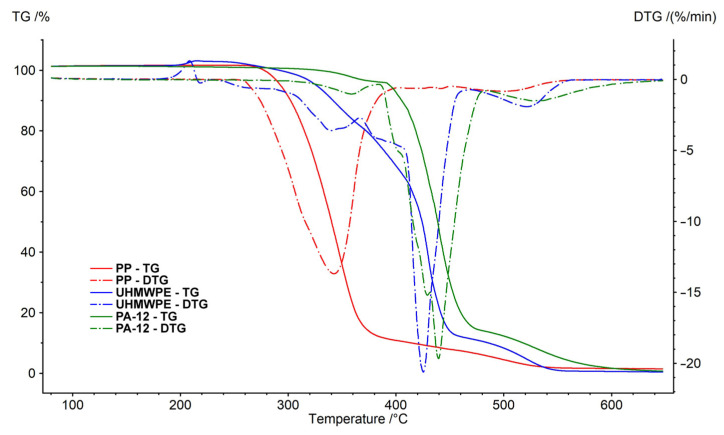
Thermogravimetric analysis of samples in air.

**Figure 9 polymers-15-04203-f009:**
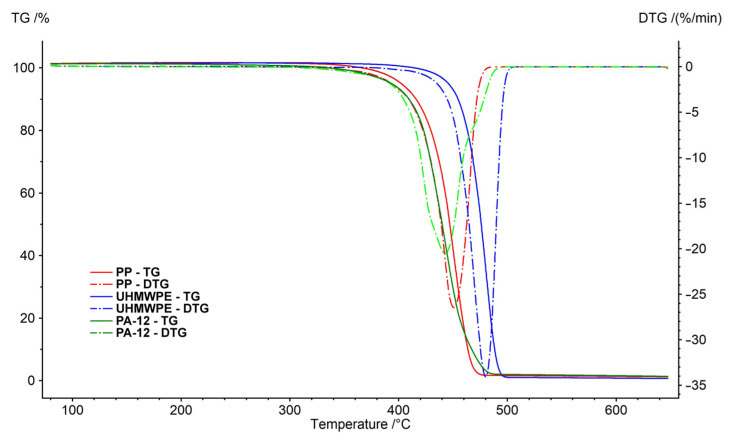
Thermogravimetric analysis of samples in nitrogen.

**Figure 10 polymers-15-04203-f010:**
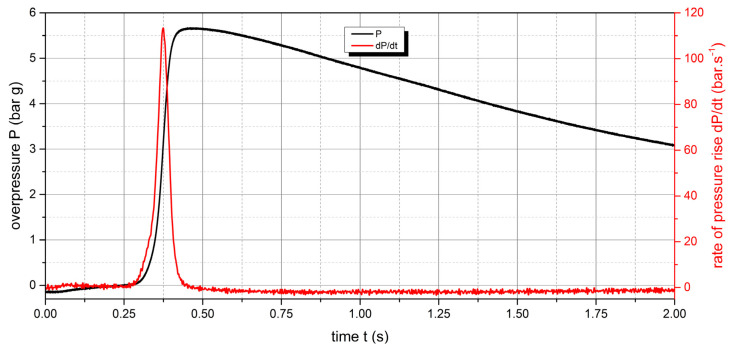
Pressure record of polyamide 12 with concentration of 250 g·m^−3^.

**Figure 11 polymers-15-04203-f011:**
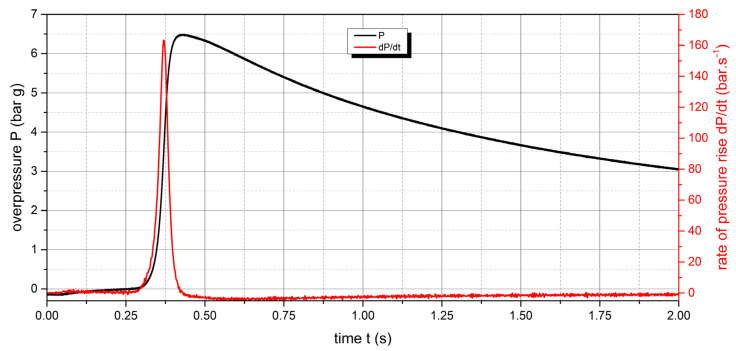
Pressure record of polyamide 12 with concentration of 500 g·m^−3^.

**Figure 12 polymers-15-04203-f012:**
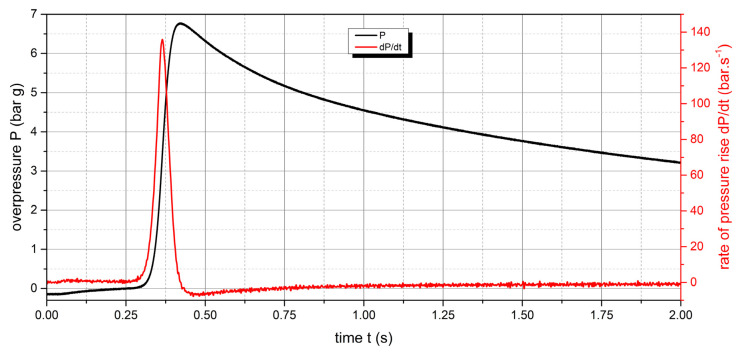
Pressure record of polyamide 12 with concentration of 750 g·m^−3^.

**Figure 13 polymers-15-04203-f013:**
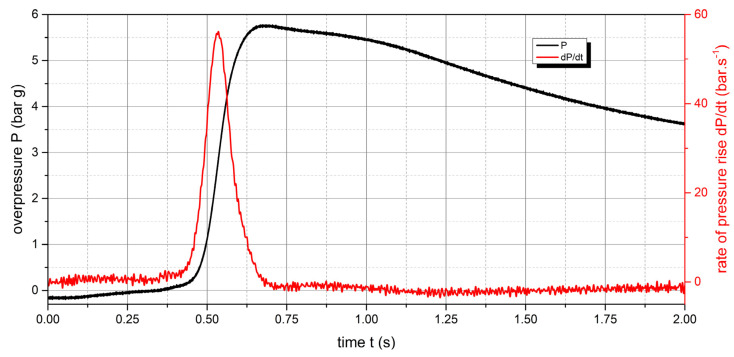
Pressure record of UHMW polyethylene with concentration of 250 g·m^−3^.

**Figure 14 polymers-15-04203-f014:**
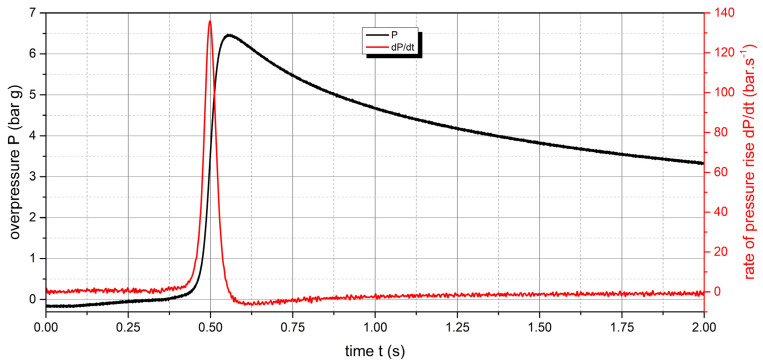
Pressure record of UHMW polyethylene with concentration of 500 g·m^−3^.

**Figure 15 polymers-15-04203-f015:**
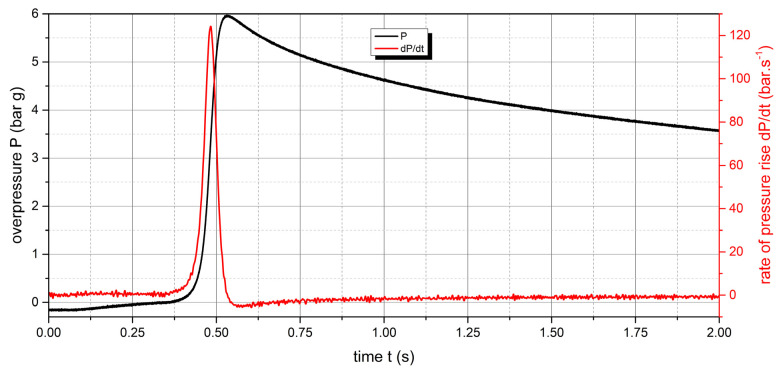
Pressure record of UHMW polyethylene with concentration of 750 g·m^−3^.

**Table 1 polymers-15-04203-t001:** Proportion of particle sizes in the samples of polyamide 12, polypropylene and UHMW polyethylene.

Sieve Size (µm)	Sample					
	Polyamide 12		UMHW Polyethylene		Polypropylene	
	% w/w	Cumulative %	% w/w	Cumulative %	% w/w	Cumulative %
500	0.08	100	32.64	100	0	100
355	0.14	99.92	29.34	67.36	1.61	100
250	0.58	99.78	21.18	38.02	8.11	98.39
180	12.11	99.2	7.42	16.84	14.86	90.28
125	43.62	87.09	6.04	9.42	19.98	75.42
90	35.12	43.47	1.47	3.38	24.09	55.44
63	7.23	8.35	1.29	1.91	17.81	31.35
45	0.91	1.12	0.53	0.62	11.62	13.54
<45	0.21	0.21	0.09	0.09	1.92	1.92
median	95 µm		293 µm		84 µm	

**Table 2 polymers-15-04203-t002:** Results of thermogravimetric measurement of samples in air.

Polymer		Polypropylene	UHMW Polyethylene	Polyamide 12
Step 1	Range (°C)	218.5–450.8	185.8–214.5	124.4–384.1
	Peak (°C)	342.4	208.5	358.7
	Weight loss (%)	92.3	−1.6	5.2
Step 2	Range (°C)	450.8–650.0	214.5–366.4	384.1–484.0
	Peak (°C)	498.9	339.8	439.5
	Weight loss (%)	6.3	21.1	81.2
Step 3	Range (°C)	-	366.4–469.1	484.0–650.0
	Peak (°C)	-	425.4	526.8
	Weight loss (%)	-	69.3	12.9
Step 4	Range (°C)	-	469.1–650.0	-
	Peak (°C)	-	521.3	-
	Weight loss (%)	-	10.8	-
Residue at 650 °C (%)		1.4	0.4	0.7

**Table 3 polymers-15-04203-t003:** Results of thermogravimetric measurements of samples in nitrogen.

Polymer		Polypropylene	UHMW Polyethylene	Polyamide 12
Step 1	Range (°C)	221.4–650.0	287.1–650.0	126.3–650.0
	Peak (°C)	450.9	480.1	442.7
	Weight loss (%)	98.8	99.3	98.7
Residue at 650 °C (%)		1.2	0.7	1.3

**Table 4 polymers-15-04203-t004:** Explosion parameters of polymer samples with measurement uncertainties.

Sample	Polyamide 12		UHMW Polyethylene		Polypropylene
Concentration (g·m^−3^)	P_max_ (bar g)	dP/dt (bar·s^−1^)	P_max_ (bar g)	dP/dt (bar·s^−1^)	
30	0.75 (0.73 ± 0.02)	5.6 (5.4 ± 0.2)	–	–	NO EXPLOSION
60	2.51 (2.49 ± 0.02)	24.5 (24.2 ± 0.3)	1.81 (1.75 ± 0.06)	12.6 (12.2 ± 0.3)	
125	4.29 (4.21 ± 0.07)	56.5 (55.0 ± 1.3)	4.14 (4.11 ± 0.03)	34.9 (33.9 ± 0.9)	
250	5.66 (5.62 ± 0.04)	113.3 (111.6 ± 1.5)	5.77 (5.73 ± 0.04)	56.2 (55.5 ± 1.0)	
500	6.48 (6.41 ± 0.07)	163.2 (161.6 ± 1.7)	6.47 (6.45 ± 0.02)	135.9 (134.5 ± 1.5)	
750	6.76 (6.65 ± 0.10)	135.8 (133.9 ± 1.7)	5.97 (5.92 ± 0.04)	124.2 (123.4 ± 0.7)	
1000	6.33 (6.27 ± 0.06)	111.5 (109.8 ± 1.6)	5.67 (5.65 ± 0.03)	87.3 (85.9 ± 1.6)	

**Table 5 polymers-15-04203-t005:** Measured values of MIT sample polyamide 12.

Sample Weight (g)	Air Pressure (kPa)	Temperature (°C)	Results
0.2	50	450	YES
		440	YES
		430	YES
		420	YES
		410	YES
		400	YES
		390	YES
		380	YES
		370	NO
0.11	50	370	NO
		380	YES
		370	NO

**Table 6 polymers-15-04203-t006:** Measured values of MIT sample UHMW polyethylene.

Sample Weight (g)	Air Pressure (kPa)	Temperature (°C)	Results
0.11	50	370	YES
		360	YES
		350	YES
		340	NO

## Data Availability

The data presented in this study are available on request from the corresponding author.

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
