# Peer review of "Study into the Fire and Explosion Characteristics of Polymer Powders Used in Engineering Production Technologies"

_polymers, 2023, doi:10.3390/polym15214203_

Round 1

Reviewer 1 Report

ABSTRACT
Authors should emphasize the key achievements unique to their study in the abstract, such as references to X-RAY, FTIR, and TGA. Explosion data is obvious and can be found in the literature and GESTIS-DUST-EX.

Line 23: Which type of polymer powder was found to be non-flammable?
Line 32: What safety measures should be used?

INTRODUCTION
The Introduction section can be enhanced by incorporating more recent works in this field, providing a comprehensive literature review of the fire and explosion characteristics of different polymer powders and the methods used to study them. The reference list is deficient. Several recent topical

references concerning the explosion research method of dust are not included, e.g. https://doi.org/10.3390/en16176121.

Additionally, the motivation and contributions of the study should be articulated more clearly.

MATERIALS AND METHODS
Lines 253-259: Sentences related to TGA and DSC should be relocated to the Materials and Methods section.

What is the volume of the chamber?

RESULTS AND DISCUSSION
In Table 4, uncertainties of explosion parameters should be included.
What about the constraints of heat and mass transfer in TGA experiments?

CONCLUSION
The conclusion should better reflect the paper's content and main findings. Furthermore, the practical implications of the results obtained in this study should be emphasized, and future research directions should be outlined.

The English language in the paper needs improvement. The authors should consider revising the manuscript to eliminate language imperfections and enhance clarity.

Author Response

Dear Reviewer, 

 Thank you so much for your letter and advice. We have revised the manuscript, and would like to re-submit it for your consideration. We have addressed the reviewers’ comments. Point by point responses to the reviewers’ comments are listed below this letter. We hope that the revised manuscript is now accepted for publication in journal.

Also, we would like to express our sincere thanks for the constructive and positive comments. 

Thank you and best regards.

Yours sincerely authors.

Comment 1: Authors should emphasize the key achievements unique to their study in the abstract, such as references to X-RAY, FTIR, and TGA. Explosion data is obvious and can be found in the literature and GESTIS-DUST-EX.

Response: Key achievements has been corrected and added to abstract.

Comment 2:  Line 23: Which type of polymer powder was found to be non-flammable? 

Response: Type of non-flammable polymer has been added to abstract.

Comment 3: Line 32: What safety measures should be used?

Response: Safety measures has been added to abstract.

Comment 4: INTRODUCTION. The Introduction section can be enhanced by incorporating more recent works in this field, providing a comprehensive literature review of the fire and explosion characteristics of different polymer powders and the methods used to study them. The reference list is deficient. Several recent topical references concerning the explosion research method of dust are not included, e.g. https://doi.org/10.3390/en16176121.

Reponse: The introduction section has been enhanced and corrected.

Comment 5:  Additionally, the motivation and contributions of the study should be articulated more clearly.

Response: The motivation and contributions of study has been described in detail.

Comment 6: MATERIALS AND METHODS. Lines 253-259: Sentences related to TGA and DSC should be relocated to the Materials and Methods section.

Response: Sentences has been relocated.

Comment 7: What is the volume of the chamber?

Response: Volume of the chamber is in rows 258-259.

Comment 8: RESULTS AND DISCUSSION. In Table 4, uncertainties of explosion parameters should be included.

Response: Uncertainties of explosion parameter are in table 4 (mean value and standard deviaton of three measurements at each concentration)

Comment 9: What about the constraints of heat and mass transfer in TGA experiments?

Response: Information about constraints of heat and mass transfer in TGA experiments has been added.

Comment 10: CONCLUSION The conclusion should better reflect the paper's content and main findings. Furthermore, the practical implications of the results obtained in this study should be emphasized, and future research directions should be outlined.

Response: The conclusion has been enhanced and practical implication of the results with future research directions has been added to conclusion.

Comments on the Quality of English Language

The English language in the paper needs improvement. The authors should consider revising the manuscript to eliminate language imperfections and enhance clarity.

Response: English language in the paper has been improved.

Reviewer 2 Report

The manuscript contains the results of a study of the possibility of ignition and explosion of powdered polymer materials in air. The results shows that the use of Polypropylene BorPlus SE523MO is safe in the entire temperature range (up to 450 °C). Polyamide PA12 and UHMW Polyethylene polymers can be safely used at the temperatures up to 300 °C. 

Comments on the manuscript

 1.     The manuscript presents the results of a study of the properties of three different polymer materials Polyamide PA 12, Polypropylene PP and Polyethylene UHMW-PE, but there is no analysis of the chemical composition of these materials. This raises a reasonable question: what specific materials were studied in the work? Especially for PP samples. The manuscript presents data from work [18] on the explosion of PP in the air: “Minimum Explosible Concentration (MEC) of PP powders was determined to be 25 g.m-3”. However, the data given in the manuscript shows the absence of an explosion of the PP. “The Polypropylene sample was evaluated as non-flammable and non-explosive.” There is no satisfactory explanation for this. What specific additives were able to prevent inflammation?

2.     There are scale markings in Figures 1 and 2, but they are hard to see.

3.     There seems to be an error in the name of Table 3 (line 294)

"Table 3 Results of thermogravimetric measurements of samples in air" - need

"Table 3 Results of thermogravimetric measurements of samples in nitrogen"

 I recommend publishing the manuscript after the minor revision.

Author Response

Dear Reviewer,

 Thank you so much for your letter and advice. We have revised the manuscript, and would like to re-submit it for your consideration. We have addressed the reviewers’ comments. Point by point responses to the reviewers’ comments are listed below this letter. We hope that the revised manuscript is now accepted for publication in journal.

Also, we would like to express our sincere thanks for the constructive and positive comments. 

Thank you and best regards.

Yours sincerely authors.

Comment 1: The manuscript presents the results of a study of the properties of three different polymer materials Polyamide PA 12, Polypropylene PP and Polyethylene UHMW-PE, but there is no analysis of the chemical composition of these materials. This raises a reasonable question: what specific materials were studied in the work? Especially for PP samples. The manuscript presents data from work [18] on the explosion of PP in the air: “Minimum Explosible Concentration (MEC) of PP powders was determined to be 25 g.m-3”. However, the data given in the manuscript shows the absence of an explosion of the PP. “The Polypropylene sample was evaluated as non-flammable and non-explosive.” There is no satisfactory explanation for this. What specific additives were able to prevent inflammation?

Response: MSDS of the materials used in the article are available. Polymer is listed as the majority component in them, additives are listed only as "additives". It is probably due to the manufacturer's trademark (in this PP, it uses a flame retardant as an additive). In the MSDS of a similar polymer such as PP (same manufacturer), additives are listed - silicates and phosphates. Their effect on fire parameters can be positive or negative (increases or decreases fire parameters) and the additive content can be very low (e.g. organic antioxidants). Detecting the presence of some innovative antioxidant can be difficult from the point of view of analysis (e.g. due to the complex structure of the antioxidant itself). In the end, it is always necessary to make a measurement in the laboratory, they cannot be made by calculation or simulation. In the article, we modified and supplemented this part for better understanding

Comment 2: There are scale markings in Figures 1 and 2, but they are hard to see.

Response: Figures 1 and 2 has been changed.

Comment 3: There seems to be an error in the name of Table 3 (line 294). "Table 3 Results of thermogravimetric measurements of samples in air" – need. "Table 3 Results of thermogravimetric measurements of samples in nitrogen"

Response: All errors has been corrected.

Round 2

Reviewer 1 Report

The authors have improved the paper that now could be accepted for publication. Congratulations!